# Multimorbidity and mortality in an older, rural black South African population cohort with high prevalence of HIV findings from the HAALSI Study

Alisha N Wade ![ORCID],[1] Collin F Payne,[2] Lisa Berkman,[1,3] Angela Chang,[4,5] F Xavier Gómez-Olivé,[1] Chodziwadziwa Kabudula,[1] Kathleen Kahn,[1,6] Joshua A Salomon,[7] Stephen Tollman,[1,6] Miles Witham ![ORCID] ,[1,8,9] Justine Davies[1,10,11]

MW and JD are joint senior authors.

For numbered affiliations see end of article.

**Correspondence to**
Professor Justine Davies;
j.davies.6@bham.ac.uk

## ABSTRACT

**Objectives** Multimorbidity is associated with mortality in high-income countries. Our objective was to investigate the relationship between multimorbidity (≥2 of the following chronic medical conditions: hypertension, diabetes, dyslipidaemia, anaemia, HIV, angina, depression, post-traumatic stress disorder, alcohol dependence) and all-cause mortality in an older, rural black South African population. We further investigated the relationship between HIV multimorbidity (HIV as part of the multimorbidity cluster) and mortality, while testing for the effect of frailty in all models.

**Design** Population cohort study.

**Setting** Agincourt subdistrict of Mpumalanga province, South Africa.

**Participants** 4455 individuals (54.7% female), aged ≥40 years (median age 61 years, IQR 52–71) and resident in the study area.

**Primary and secondary outcome measures** The primary outcome measure was time to death and the secondary outcome measure was likelihood of death within 2 years of the initial study visit. Mortality was determined during annual population surveillance updates.

**Results** 3157 individuals (70.9%) had multimorbidity; 29% of these had HIV. In models adjusted for age and sociodemographic factors, multimorbidity was associated with greater risk of death (women: HR 1.72; 95% CI: 1.18 to 2.50; men: HR 1.46; 95% CI: 1.09 to 1.95) and greater odds of dying within 2 years (women: OR 2.34; 95% CI: 1.32 to 4.16; men: OR 1.51; 95% CI: 1.02 to 2.24). HIV multimorbidity was associated with increased risk of death compared with non-HIV multimorbidity in men (HR 1.93; 95% CI: 1.05 to 3.54), but was not statistically significant in women (HR 1.85; 95% CI: 0.85 to 4.04); when detectable, HIV viral loads were higher in men (p=0.021). Further adjustment for frailty slightly attenuated the associations between multimorbidity and mortality risk (women: HR 1.55; 95% CI: 1.06 to 2.26; men: HR 1.36; 95% CI: 1.01 to 1.82), but slightly increased associations between HIV multimorbidity and mortality risk.

**Conclusions** Multimorbidity is associated with mortality in this older black South African population. Health systems which currently focus on HIV should be reorganised to optimise identification and management of other prevalent chronic diseases.

## Strengths and limitations of this study

► This is, to our knowledge, the first study of multimorbidity and mortality in sub-Saharan Africa and the first to compare mortality between those with HIV multimorbidity and those with non-HIV multimorbidity in any setting.
► Multimorbidity was determined using standardised self-report and objective measures.
► The study was conducted in a large populationbased cohort with ascertainment of mortality outcome data in over 99% of participants.
► The clinical severity of morbidities was not assessed and more severe morbidities might be expected to have a greater impact on mortality.
► There was a higher proportion of frailty in the individuals who were excluded due to missing data which may have diluted the impact of frailty on the relationship between multimorbidity and death.

## INTRODUCTION

Multimorbidity, commonly defined as the presence of two or more chronic conditions,[1 2] has become an increasingly important part of the public health research agenda in sub-Saharan Africa.[3–5] As the population ages, chronic non-communicable diseases such as hypertension, diabetes and ischaemic heart disease have become more common.[6] This trend is influenced by the increase in obesity linked to sedentary lifestyles and consumption of high-calorie convenience foods. As in high-income countries, these chronic conditions rarely occur in isolation, with estimates of non-communicable disease multimorbidity varying between 23% and 69% in population-based studies in older adults in sub-Saharan African countries.[3 5 7 8] Multimorbidity in many sub-Saharan African countries is, however, distinguished from that seen in high-income settings by the high prevalence

of HIV co-occurring with non-communicable diseases and younger age of onset,[9–11] with potential implications for patient outcomes and the way in which multimorbidity is managed by the healthcare system.

Although evidence from high-income countries suggests that multimorbidity is associated with a higher risk of mortality in older adults,[12] it is unclear if this holds true in sub-Saharan Africa where disease patterns and interactions with other ageing syndromes may differ, influenced by the complex health transitions underway in this region.[6] A multimorbidity disease cluster that includes HIV, even if treated, may, for example, confer a higher risk of death than one that does not, given the chronic inflammation and treatment toxicities associated with HIV. Conversely, individuals with HIV may have more frequent contact with the healthcare system and consequently have better control of coexisting non-communicable conditions.[13] Frailty, an ageing syndrome characterised by the progressive loss of ability to withstand physiological stressors,[14] appears to be more prevalent in sub-Saharan Africa than in many high-income countries.[15] Although it is associated with both multimorbidity and mortality in older African populations,[16] it is unknown whether frailty affects the relationship between multimorbidity and mortality in these populations or whether it predicts additional mortality risk beyond that predicted by multimorbidity alone.

Our objective was to investigate the relationship between multimorbidity and all-cause mortality risk in older black adults in a rural South African community. We defined older adults as those 40 years and above, given that phenotypic characteristics associated with ageing can occur in sub-Saharan Africa at younger ages than in high-income countries[16 17] and the lower life expectancy in many countries in this region. We hypothesised that multimorbidity was associated with all-cause mortality and performed exploratory subgroup analyses to investigate whether mortality risk differed when HIV was a component of the multimorbidity cluster. We further explored whether frailty modified the ability of multimorbidity to predict death.

## METHODS
### Study setting and sample
The study used data from Health and Ageing in Africa—a Longitudinal Study in an INDEPTH community (HAALSI) which has previously been described in detail.[18] In brief, HAALSI is a longitudinal cohort study which recruited individuals ≥40 years and permanently resident in the Agincourt Health and socio-Demographic Surveillance System (HDSS) in the Agincourt subdistrict of rural Mpumalanga, northeast South Africa.[19] The Agincourt HDSS is hosted by the Medical Research Council/Wits Rural Public Health and Health Transitions Research Unit which annually enumerates its population and documents all births, deaths and migrations, consequently ensuring robust denominators. Of the 12 875

eligible individuals, 6281 were randomly selected and 5059 of these individuals enrolled in the study cohort.

### Study visits
Baseline study visits took place between November 2014 and November 2015. Trained fieldworkers visited participants in their homes and collected data on sociodemographic variables and self-reported health status and risk factors using Computer-Assisted Personal Interviews (CAPI) and performed clinical assessments including blood pressure, measures of physical performance and point-of-care biomarkers. Dried blood spots were also collected for assessment of HIV serostatus and viral load. The survey instruments were translated from English into Shangaan, the local language, and responses were back-translated into English to ensure reliability. Translation was performed by experienced members of the unit staff with further minor modifications by the fieldworkers who conducted the interviews to ensure the language used was in keeping with the vernacular.

### Definition of variables
#### Sociodemographic variables
Sociodemographic variables were self-reported and included age (as a continuous variable and calculated as the difference between the date of birth and date of interview), marital status (classified as never married, previously married and currently married) and educational attainment (classified as no formal education, at least some primary education, at least some secondary education and completed secondary education or more). Principal component analysis was used to create a wealth index from household characteristics and asset ownership.[20]

#### Multimorbidity
Participants were considered to have multimorbidity if they had two or more[1 2] of the chronic illnesses listed below.[3] We further classified the subset of individuals who had HIV as one of their chronic conditions as having HIV multimorbidity. The chronic illnesses included were selected to ensure comparability with Health and Retirement Survey sister studies as well as to obtain further data on conditions which are prevalent in the study area.[18]

#### Chronic illnesses
The 10 included chronic illnesses were defined by a combination of self-report and objective measures.[3] Individuals who reported existing diagnoses of hypertension, diabetes and dyslipidaemia were classified as having these conditions. In addition, the following objective measures were used for classification: hypertension—mean systolic blood pressure ≥140 mm Hg or mean diastolic blood pressure ≥90 mm Hg calculated from the second and third of three consecutive measurements during the home visit[21]; diabetes mellitus—fasting glucose ≥7 mmol/L or random glucose ≥11.1 mmol/L on point-of-care testing during the home visit (CareSens N monitor; i-SENS, Seoul, South Korea)[22] and dyslipidaemia—total cholesterol >6.21

mmol/L, high-density lipoprotein <1.19 mmol/L, low-density lipoprotein >4.1 mmol/L or triglycerides >2.25 mmol/L on point-of-care testing (CardioChek PA; PTS Diagnostics, Whitestown, Indiana, USA).[3] Anaemia was defined as haemoglobin <120 g/L in women and <130 g/L in men on point-of-care testing (Hemocue Hb 201+analyser; Haemocue, Sweden).[23] Individuals were considered HIV positive if dried blood spots were positive on screening (Vironostika Uniform 11; Biomeriuex, France) and subsequent confirmatory tests (Roche Elecsys; Roche, USA). Dried blood spots from individuals who tested positive for HIV were then tested for HIV-1 RNA (BioMérieux NucliSens; lower limit of detection 100 copies/mL).

Angina was defined using the Rose criteria[24] and chronic bronchitis was defined as a self-reported daily cough, productive of phlegm, for at least 3 months per year for at least 2 successive years.[25] Participants were classified as having depression if they identified three or more symptoms on the Centre for Epidemiological Studies-Depression (CES-D) Scale,[26] while post-traumatic stress disorder was defined as a score ≥4 on the Breslau Scale.[27] Individuals were considered to have alcohol dependence if they gave affirmative answers to two or more questions on the CAGE screening questionnaire.[28]

### Physical function and determination of frailty

We assessed phenotypic frailty status using a locally adapted version of Fried's frailty criteria,[29] which has demonstrated construct and predictive validity.[16] Participants scored 1 point for each of slow walking speed (lowest gait speed quintile (0.46 m/s)), low muscle strength (lowest quintile of grip strength (<21.6 kg in men and <16.4 kg in women) assessed using a baseline digital Smedley hand dynamometer (Fabrication Enterprises, Hong Kong)), body mass index <18.5 kg/m$^2$, low activity level (lowest quintile for weekly metabolic equivalents calculated from the Global Physical Activity Questionnaire[30] (<30 metabolic equivalents/week in men and <50 metabolic equivalents/week in women)) and self-reported exhaustion based on a positive response to the question 'much of the time you could not get going' from the CES-D Scale. Individuals with a total score of '0' were classified as non-frail, those with a total of '1' or '2' as pre-frail and those with a total of '3' or more or those who were physically unable to complete a component of the assessment were classified as frail.

### Determination of outcome measures

Our primary outcome was time to death, while we secondarily investigated likelihood of death within 2 years of the initial study visit. All-cause mortality was determined during the annual HDSS census update rounds[19] and the date of death was ascertained through an interview with a household member of the decedent.

### Statistical analysis

Continuous variables were not normally distributed and were therefore summarised using medians and IQRs while categorical variables were summarised using percentages. Mann-Whitney and X$^2$ tests were used to compare continuous and categorical variables, respectively, between groups defined by inclusion/exclusion in the analysis, sex and multimorbidity status.

We constructed Cox proportional hazards models to investigate the relationship between age-adjusted and sex-adjusted multimorbidity status and risk of death in the entire sample and in age-adjusted models stratified by sex. Models were further sequentially adjusted for sociodemographic variables (marital status, educational attainment and wealth quintile) and frailty status, variables which were associated with mortality in previous research in this population.[16 31] Age–sex interaction was statistically significant and was therefore included in models of the entire sample. Individuals were censored at the date of their 2018 census visit or at the date of permanent out-migration from the study area. Any individuals lost to follow-up prior to the 2018 census round were censored at the date of their last census visit.

Similar models were constructed to explore the relationship between HIV multimorbidity and risk of death in the subgroup of individuals with multimorbidity. The interaction between HIV multimorbidity and time was statistically significant and was therefore included in the models. Cox models satisfied the assumption of proportional hazards using Schoenfeld residual plots.

In supplementary analyses, logistic regression models were used to investigate the association between age-adjusted and sex-adjusted multimorbidity (both any multimorbidity and HIV multimorbidity) and mortality within 2 years in the entire sample and age-adjusted multimorbidity and 2-year mortality in models stratified by sex. Models were sequentially adjusted as in the Cox proportional hazards models and the areas under the curve were calculated for each model.

Sensitivity analyses of the primary outcome were conducted in which the chronic conditions with the highest degree of missingness were excluded from the definition of multimorbidity.

P<0.05 was considered statistically significant. Analyses were performed using STATA V.14.2 (StataCorp, USA).

### Patient and public involvement

Prior to the initiation of the HAALSI Study, an extensive process of community engagement was led by the Agincourt Office of Public Engagement. This included meetings with the Community Advisory Group, nominated by Community Development Forums and civic and traditional leadership structures to discuss planned research activities. Feedback on the results of this study will be included in the annual feedback of study results to villagers and community leaders.

## RESULTS

Six hundred and four (11.9%) cohort members were excluded from this analysis due to missing data, the

**Table 1** Sociodemographic, frailty and mortality characteristics overall and by morbidity status

| | Total (n=4455) | No multimorbidity (n=1298; 29.1%) | Multimorbidity (n=3157; 70.9%) | P value | Non-HIV multimorbidity (n=2242; 71.0%) | HIV multimorbidity (n=915; 29.0%) | P value |
|---|---|---|---|---|---|---|---|
| Age (years) | 61 (52–71) | 60 (50–70) | 61 (52–72) | <0.001 | 64 (55–74) | 55 (48–62) | <0.001 |
| Female (%) | 2437 (54.7) | 667 (51.4) | 1770 (56.1) | 0.004 | 1261 (56.2) | 509 (55.6) | 0.752 |
| Marital status (%) | | | | <0.001 | | | <0.001 |
| Never married | 226 (5.1) | 80 (6.2) | 146 (4.6) | | 86 (3.8) | 60 (6.6) | |
| Previously married | 1934 (43.4) | 467 (36.0) | 1467 (46.5) | | 967 (43.1) | 500 (54.6) | |
| Currently married | 2295 (51.5) | 751 (57.9) | 1544 (48.9) | | 1189 (53.0) | 355 (38.8) | |
| Educational attainment (%) | | | | 0.008 | | | <0.001 |
| No formal education | 2036 (45.7) | 562 (43.3) | 1474 (46.7) | | 1092 (48.7) | 382 (41.8) | |
| Some primary education | 1550 (34.8) | 444 (34.2) | 1106 (35.0) | | 784 (35.0) | 322 (35.2) | |
| Some secondary education | 505 (11.3) | 163 (12.6) | 342 (10.8) | | 201 (9.0) | 141 (15.4) | |
| Secondary or more | 364 (8.2) | 129 (9.9) | 235 (7.4) | | 165 (7.4) | 70 (7.7) | |
| Wealth index quintile (%) | | | | 0.868 | | | <0.001 |
| 1st | 915 (20.5) | 273 (21.0) | 642 (20.3) | | 417 (18.6) | 225 (24.6) | |
| 2nd | 887 (199) | 257 (19.8) | 630 (20.0 | | 438 (19.5) | 192 (21.0) | |
| 3rd | 878 (19.7) | 264 (20.3) | 614 (19.5) | | 429 (19.1) | 185 (20.2) | |
| 4th | 881 (19.8) | 246 (19.0) | 635 (20.1) | | 465 (20.7) | 170 (18.6) | |
| 5th | 894 (20.1) | 258 (19.9) | 636 (20.2) | | 493 (22.0) | 143 (15.6) | |
| Frailty status (%) | | | | <0.001 | | | <0.001 |
| Non-frail | 2347 (52.7) | 811 (62.5) | 1536 (48.7) | | 1003 (44.7) | 533 (58.3) | |
| Pre-frail | 1648 (37.0) | 398 (30.7) | 1250 (39.6) | | 925 (41.3) | 325 (35.5) | |
| Frail | 460 (10.3) | 89 (6.9) | 371 (11.8) | | 314 (14.0) | 57 (6.2) | |
| Undetectable viral load* (%) | | | | | | 475 (51.9) | |
| Viral load (copies/mL) | | | | | | 2200 (630–18 000) | |

*Two individuals with HIV had insufficient sample for viral load testing. Undetectable viral load was defined as <100 copies/mL. Continuous variables are summarised as medians and IQRs and compared using the Mann-Whitney test. Categorical variables were compared using the $X^2$ test.

majority of whom (94.7%) were missing data on multimorbidity status. Missing multimorbidity status was primarily due to missing data on dyslipidaemia and anaemia which resulted from occasional malfunctioning of the measurement device in high-ambient temperatures, rather than selection bias. There was no difference in age between those who were included and excluded (p=0.295), while slightly more men were excluded (13.9% vs 10.2%; p<0.001). Frailty distribution differed between the two groups (p<0.001), with a greater proportion of frail individuals among those excluded (39.2% vs 10.3%). Three of the 4455 individuals in the study sample were lost to follow-up prior to their 2-year study visit; data were available on all other members of the study sample for a minimum of 2 years after the baseline study visit.

Multimorbidity was present in 3157 individuals (70.9%) in the study sample at the time of the baseline study visit (table 1). People with multimorbidity were slightly older than those without (median age 61 vs 60 years; p<0.001)

and there were more women than men in the multimorbidity group (56.1% vs 51.4%; p=0.004). Multimorbidity was associated with frailty status, with more individuals with multimorbidity in the pre-frail and frail categories (p<0.001). Nine hundred and fifteen (29%) of the 3157 individuals with multimorbidity had HIV as one of their multimorbid conditions. Those with HIV multimorbidity were younger (median age 55 vs 64 years; p<0.001) and less frail than those with non-HIV multimorbidity (p<0.001). Just over half of those with HIV multimorbidity had an undetectable viral load. There was no difference in the proportion of women and men with HIV multimorbidity who had an undetectable viral load (52.9% vs 50.7%; p=0.811), but men with HIV multimorbidity and a detectable viral load had a higher median viral load than women in the same category (3700 copies/mL (IQR 790–24 000) vs 1600 copies/mL (IQR 480–16 000); p=0.021). While hypertension was the most prevalent individual disease in those with multimorbidity (75.9%),

**Table 2** Characteristics of participants who died during the follow-up period

| | n=459 |
|---|---|
| Age (years) | 71 (60–81) |
| Female (%) | 200 (43.6) |
| Marital status (%) | |
| Never married | 22 (4.8) |
| Previously married | 246 (53.6) |
| Currently married | 191 (41.6) |
| Educational attainment (%) | |
| No formal education | 278 (60.6) |
| Some primary education | 142 (30.9) |
| Some secondary education | 31 (6.8) |
| Secondary or more | 8 (1.7) |
| Wealth index quintile (%) | |
| 1st | 102 (22.2) |
| 2nd | 93 (20.3) |
| 3rd | 100 (21.8) |
| 4th | 85 (18.5) |
| 5th | 79 (17.2) |
| Frailty status (%) | |
| Non-frail | 141 (30.7) |
| Pre-frail | 187 (40.7) |
| Frail | 131 (28.5) |
| Multimorbidity (%) | 367 (80.0) |
| HIV multimorbidity (%) | 88 (24.0) |
| Undetectable viral load (%) | 34 (38.6) |
| Viral load (copies/mL) | 18 000 (1700–37 000) |

Undetectable viral load was defined as <100 copies/mL.
Continuous variables are summarised as medians and IQRs; HIV multimorbidity is presented as a subgroup of those with multimorbidity and data on viral load are for the subgroup with HIV multimorbidity.

the frequency of individual conditions varied by sex, age group and HIV multimorbidity status; hypertension was the most frequent condition in those with non-HIV multimorbidity, for example, but the second most frequent

in those with HIV multimorbidity (online supplemental table 1).

Median follow-up time was 3.3 years (IQR 3.2–3.5 years), during which time 459 people died. The median age of those who died was 71 years (IQR 60–81) and 200 (43.6%) of these were women. Eighty per cent of those who died had multimorbidity and of those, 88 (24.0%) had HIV multimorbidity. Approximately 39% of those with HIV multimorbidity who died had an undetectable viral load (table 2).

After adjustment for age and sociodemographic factors, multimorbidity, compared with no multimorbidity, was associated with greater risk of death in both women (HR 1.72; 95% CI: 1.18 to 2.50) and men (HR 1.46; 95% CI: 1.09 to 1.95) (table 3). The results remained significant after further adjustment for frailty status (women HR 1.55; 95% CI: 1.06 to 2.26 and men HR 1.36; 95% CI: 1.01 to 1.82). HIV multimorbidity, compared with non-HIV multimorbidity and similarly adjusted for age and sociodemographic factors, was associated with an increased risk of death in men (HR 1.93; 95% CI: 1.05 to 3.54), with a non-statistically significant association in women (HR 1.85; 95% CI: 0.85 to 4.04). HIV multimorbidity was also associated with a significantly increased risk of death for both men and women after further adjustment for frailty status.

Results of the odds of death within 2 years of the data collection are presented in the online supplemental files. In summary, in a model fully adjusted for age, sociodemographic factors and frailty, there was a 58% increase in the odds of death within 2 years in individuals with multimorbidity compared with those without multimorbidity (OR 1.58; 95% CI: 1.14 to 2.19) (online supplemental table 2). The model was able to correctly discriminate between those who did not die within 2 years and those who did in 76.7% (95% CI: 73.9% to 79.5%) of the pairs considered, with the addition of frailty resulting in a small but significant improvement in discriminant ability over the model in which multimorbidity was adjusted only for age and sociodemographic factors (figure 1).

In fully adjusted models stratified by sex, women with multimorbidity were twice as likely to be dead within 2 years of the baseline visit as those without multimorbidity (OR 2.05; 95% CI: 1.14 to 3.69); there was no statistically

**Table 3** HRs for risk of death by multimorbidity status

| | Any multimorbidity (HR (95% CI)) | | | HIV multimorbidity* (HR (95% CI)) | | |
|---|---|---|---|---|---|---|
| | Total† (n=4455) | Women (n=2437) | Men (n=2018) | Total† (n=3157) | Women (n=1770) | Men (n=1387) |
| Model 1 | 1.56 (1.24 to 1.97) | 1.72 (1.18 to 2.49) | 1.47 (1.10 to 1.97) | 2.08 (1.30 to 3.35) | 1.97 (0.91 to 4.26) | 2.10 (1.15 to 3.84) |
| Model 2 | 1.55 (1.23 to 1.95) | 1.72 (1.18 to 2.50) | 1.46 (1.09 to 1.95) | 1.92 (1.19 to 3.10) | 1.85 (0.85 to 4.04) | 1.93 (1.05 to 3.54) |
| Model 3 | 1.42 (1.13 to 1.79) | 1.55 (1.06 to 2.26) | 1.36 (1.01 to 1.82) | 2.59 (1.57 to 4.27) | 2.75 (1.19 to 6.37) | 2.34 (1.26 to 4.34) |

Model 1: adjusted for age.
Model 2: adjusted for age, marital status, education status and socioeconomic status.
Model 3: adjusted for age, marital status, education status, socioeconomic status and frailty.
*Adjusted for HIV multimorbidity–time interaction.
†Adjusted for sex and age–sex interaction term.

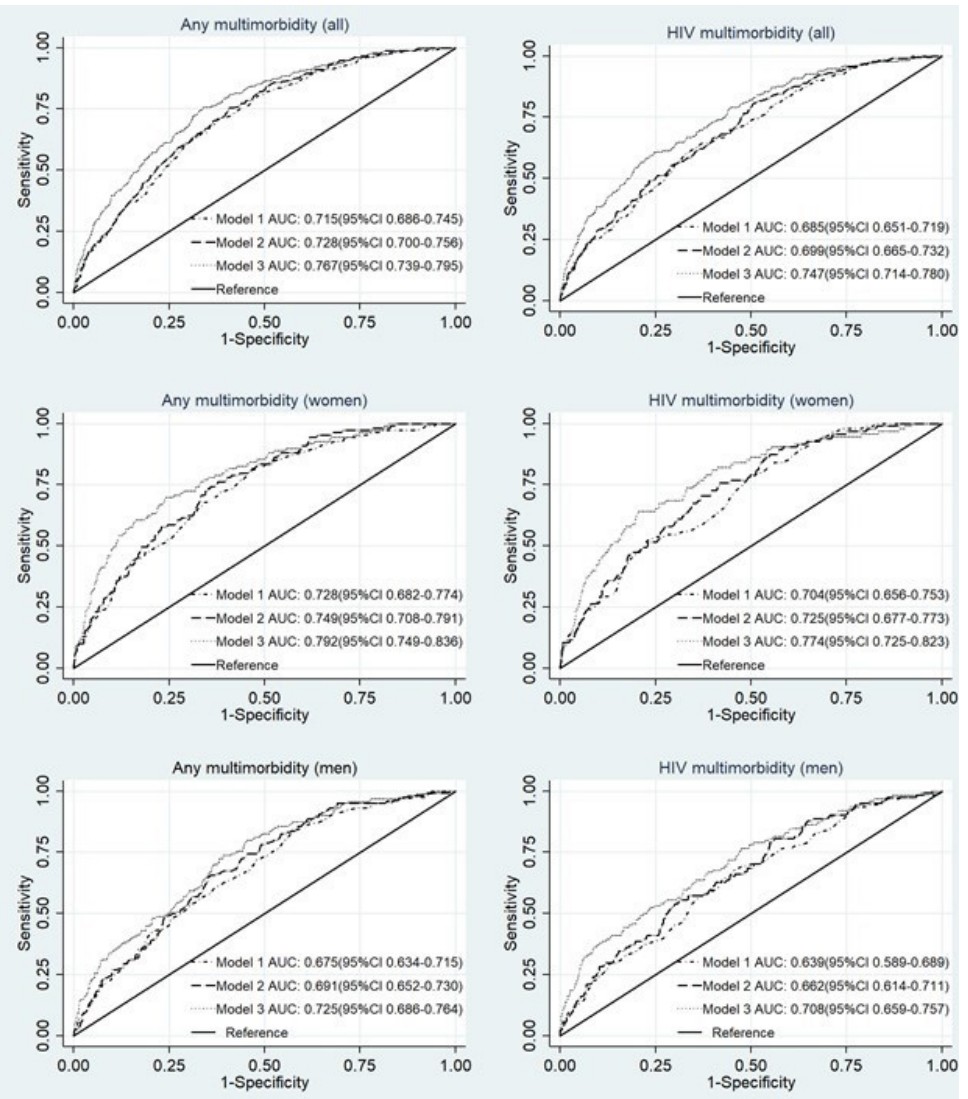

**Figure 1** Receiver operating curves for multimorbidity and death within 2 years. Model 1: any multimorbidity/HIV multimorbidity and age; model 2: any multimorbidity/HIV multimorbidity, marital status, educational attainment and socioeconomic status; model 3: any multimorbidity/HIV multimorbidity, marital status, educational attainment, socioeconomic status and frailty status. Models constructed for total sample were additionally adjusted for sex. AUC, area under the curve.

significant association in men (OR 1.38; 95% CI: 0.92 to 2.05). The model was able to correctly discriminate between those women who did not die within 2 years and those who did in 79.2% (95% CI: 74.9% to 83.6%) of pairs, while in men it correctly discriminated in 72.5% (95% CI: 68.6% to 76.4%) of cases (figure 1). The addition of frailty resulted in a marginal improvement, in women, in discriminant ability over the model in which multimorbidity was adjusted only for age and sociodemographic factors.

In similarly fully adjusted models, there was a 60% increase in the odds of death within 2 years in individuals with HIV multimorbidity compared with those with non-HIV multimorbidity (OR 1.60; 95% CI: 1.12 to 2.30) (online supplemental table 3). Adjustment of HIV multimorbidity for frailty, in addition to age and sociodemographic factors, significantly improved the discriminant ability of the model (figure 1). In sex-stratified models,

there was a 64% increase in the odds of death within 2 years in men with HIV multimorbidity compared with those with non-HIV multimorbidity (OR 1.64; 95% CI: 1.03 to 2.61), while in women, the association was not statistically significant. As with any multimorbidity, models that included frailty marginally improved, in women, on the discriminant ability of models in which HIV multimorbidity was adjusted only for age and sociodemographic factors.

In sensitivity analyses, dyslipidaemia and anaemia, the most frequently missing morbidities, were excluded from the chronic conditions that contributed to the definition of multimorbidity. This resulted in the exclusion of fewer cohort members and there was a final sample size of 4907 individuals, of whom 1719 (35%) had multimorbidity; 637 (37.1%) of these had HIV multimorbidity. Multimorbidity remained significantly associated with a higher risk of death in both women (HR 1.41; 95% CI: 1.07 to 1.86)

and men (HR 1.53; 95% CI: 1.21 to 1.95) after adjustment for age, sociodemographic factors and frailty. Adjusted HIV multimorbidity, when compared with non-HIV multimorbidity, was associated with a statistically significant marginally higher risk of death in men only (HR 2.13; 95% CI: 1.00 to 4.57) (online supplemental table 4).

## DISCUSSION

In this older, rural, black South African cohort, multimorbidity was associated with greater risk of death in both women and men. Men who had HIV as part of their multimorbidity cluster had a higher risk of death than those with non-HIV multimorbidity; the lack of a statistically significant increased risk in women may be because we were underpowered to detect this. The relationship between multimorbidity and death remained robust after adjusting for a range of sociodemographic variables as well as after adjusting for frailty status, itself a strong independent risk factor for mortality. Addition of frailty to models of multimorbidity adjusted for age and other sociodemographic factors slightly increased the ability of these models to discriminate between those who died within 2 years and those who did not.

Although there is a paucity of literature on multimorbidity and mortality in sub-Saharan Africa, our results are in keeping with similar population-based studies in high-income countries, where several studies report associations between multimorbidity and mortality, and the few studies done in other middle-income countries. One-year mortality rate ratios for individuals ≥40 years in a Canadian population ranged between 2 and 6,[32] although in this study, multimorbidity was defined more conservatively as three or more chronic conditions, rather than two. In a Chinese population aged 45 years and older, multimorbidity was associated with a 95% increase in risk of death during a 4-year follow-up,[33] while in Brazilian adults ≥60 years, risk for all-cause mortality over 16 years in individuals with two or more chronic diseases ranged between 36% and 105% in women and 58%–59% in men.[34] In contrast, an insurance claims-based study in Germans aged 65 years and older identified only a small association between multimorbidity and mortality during a 5-year follow-up period.[35] This study, however, defined multimorbidity as 2 or more of 46 conditions, some of which were not chronic and would also be expected to have a relatively small effect on mortality, which was reflected in the range of HRs of the individual conditions in the study (0.70–2.29 in women and 0.66–1.95 in men).

This is, to our knowledge, the first study of the relationship between multimorbidity and mortality in sub-Saharan Africa, a region experiencing rapid health transition and with an ageing population and increasing prevalence of multimorbidity. It is also, again to our knowledge, the first study to compare mortality in those with HIV as part of their multimorbidity cluster to those with non-HIV multimorbidity. The study was conducted in a large population-based cohort, using standardised self-reported and objectively assessed morbidities. Our follow-up was also robust, with ascertainment of mortality outcome data in over 99% of participants.

Our study was, however, limited by a number of factors. We used the definition of multimorbidity proposed by the Academy of Medical Sciences,[2] namely of the co-occurrence of two or more chronic conditions. This definition is primarily concerned with the effects of having more than one chronic condition and does not specify which chronic conditions should be considered nor does it give primacy to one type of chronic condition over another. The association between multimorbidity and mortality might therefore differ, depending on the prognoses of the conditions used when defining multimorbidity. Additionally, this definition of multimorbidity does not consider the clinical severity or treatment status of the constituent morbidities, other factors which would also be expected to impact mortality. We also considered multimorbidity as a homogeneous concept, and there is some evidence that multimorbidity clusters, defined by groupings of similar diseases,[36] are associated with differential mortality risk.[37] Our use of a modified version of the Fried criteria to define frailty, in which we used a threshold body mass index of 18.5 kg/m$^2$ to define unintentional weight loss, may have increased the prevalence of frailty. Our adoption of the population-specific lowest quintile to identify those with low walk speed, grip strength and physical activity was, however, in keeping with the approach used by Fried and colleagues and our context-specific frailty criteria have, as previously noted, demonstrated construct and predictive validity in this setting, Lastly, we did note a higher proportion of frailty in individuals who were excluded due to missing data which may have diluted the impact of frailty on the relationship between multimorbidity and death. However, in sensitivity analyses in which the most frequently missing individual conditions were omitted from the definition of multimorbidity, the relationship between multimorbidity and mortality persisted.

Our study has implications for both clinical care and health policy in South Africa and other sub-Saharan African countries undergoing health transitions with similar demographics and disease patterns. Clinicians should be aware that the presence of multimorbidity may increase the risk of death in older adults in the short term, even in the absence of impairments in physical function. Aggressive screening and risk factor modification should therefore be employed in individuals with single morbidities who are at high risk of developing a second condition. HIV multimorbidity was associated with higher risk of death despite those with HIV multimorbidity being younger and less frail than those without it. This may be due, in part, to only half of those with HIV multimorbidity having an undetectable viral load, with an even lower proportion in those who died, highlighting the need for improved HIV control in the population. The apparent sex differential in the relationship between HIV multimorbidity and mortality merits further discussion. There were fewer deaths among women and our study may therefore have been underpowered to detect a statistically significant association between HIV multimorbidity and mortality in this group. There are also likely other confounders of the relationship between HIV multimorbidity and mortality, such as risk-taking behaviour, which may differ between sexes

and were not included in our models. However, when men with HIV had detectable viral loads, these were higher than in women, which may reflect differences in how women and men access HIV care and adhere to treatment and consequently influence mortality.[38 39] Testing and treatment initiation and adherence campaigns may therefore need to target men more specifically. Our finding that frailty status only modestly enhances the ability of multimorbidity, adjusted for age and sociodemographic factors, to discriminate between those who did and did not die within 2 years suggests that frailty assessments may be most useful in assessing mortality risk in older adults without multimorbidity.

Our results also have implications for the way in which health systems are structured in South Africa. HIV care frequently serves as the entry point into chronic healthcare for older adults and adherence to HIV treatment programmes may be associated with improvements in non-communicable disease outcomes.[40] Our findings suggest, however, a significant need to broaden the scope of chronic care beyond HIV and ensure that health services are equipped to detect and manage a wide range of chronic conditions in community-dwelling older adults.

## CONCLUSION

Our study indicates greater risk of all-cause mortality in older adults with multimorbidity in a sub-Saharan African country. Further research is, however, required to investigate the relationship in this population between mortality and multimorbidity disease clusters as well as the clinical severity of multimorbid conditions. In the interim, models of care which optimise identification and management of multimorbidity, regardless of HIV status, need to be developed.[41]

**Author affiliations**
[1]MRC/Wits Rural Public Health and Health Transitions Research Unit, School of Public Health, University of the Witwatersrand Faculty of Health Sciences, Johannesburg, Gauteng, South Africa
[2]School of Demography, Research School of Social Sciences, Australian National University, Canberra, Australian Capital Territory, Australia
[3]Harvard Centre for Population and Development Studies, Harvard University T H Chan School of Public Health, Cambridge, Massachusetts, USA
[4]Department of Clinical Research, University of Southern Denmark, Odense, Denmark
[5]Danish Institute for Advanced Study, University of Southern Denmark, Odense, Denmark
[6]Umeå Centre for Global Health Research, Umeå University, Umeå, Sweden
[7]Department of Health Policy, Stanford University School of Medicine, Palo Alto, California, USA
[8]AGE Research Group, NIHR Newcastle Biomedical Research Centre, Translational Clinical Research Institute, Newcastle University, Newcastle upon Tyne, UK
[9]Newcastle upon Tyne Hospitals NHS Foundation Trust, Newcastle upon Tyne, UK
[10]Institute of Applied Health Research, University of Birmingham, Birmingham, UK
[11]Centre for Global Surgery, Department of Global Health, Stellenbosch University, Stellenbosch, Western Cape, South Africa

**Acknowledgements** We would like to acknowledge the residents of the Agincourt subdistrict and Health and socio-Demographic Surveillance System who participated in this study. MW acknowledges support from the NIHR Newcastle Biomedical Research Centre.

**Contributors** ANW, LB, FXG-O, CK, KK, JAS and ST were involved in design and data collection for HAALSI. ST, KK, CK and FXG-O established the HDSS and mortality data collection system. ANW, MW and JD designed this study and ANW performed the statistical analysis. ANW wrote the first draft of the manuscript. ANW, CFP, LB, AC, FXG-O, CK, KK, JAS, ST, MW and JD critically revised the manuscript and approved the final version.

**Funding** This work was supported by the National Institute on Ageing at the National Institutes of Health (grant number 1P01AG041710-01A1); the Department of Science and Innovation, South Africa (no grant number); the University of the Witwatersrand (no grant number); and the Medical Research Council (MRC), South Africa (no grant number); and previously the Wellcome Trust, UK (grant numbers 058893/Z/99/A; 069683/Z/02/Z; 085477/Z/08/Z; 085477/B/08/Z) to the MRC/Wits Rural Public Health and Health Transitions Research Unit and Agincourt Health and socio-Demographic Surveillance System, a node of the South African Population Research Infrastructure Network (SAPRIN); the Fogarty International Centre of the National Institutes of Health (grant number K43TW010698) to ANW; the Australian Research Council Discovery Early Career Award (DE210100087) and the Australian National University Futures Scheme (no grant number) to CFP.

**Competing interests** None declared.

**Patient consent for publication** Not required.

**Ethics approval** Informed consent was obtained from all participants in Shangaan, the local language. Ethical approval was obtained from the Human Research Ethics Committee (Medical) of the University of the Witwatersrand (ref. M141159), Institutional Review Board of the Harvard T H Chan School of Public Health, Office of Human Research Administration (ref. C13-1608-02) and Mpumalanga Province Research and Ethics Committee. The research conformed to the principles in the Declaration of Helsinki.

**Provenance and peer review** Not commissioned; externally peer reviewed.

**Data availability statement** Data are available in a public, open access repository. Data are available upon reasonable request. The HAALSI baseline data are publicly available at the Harvard Centre for Population and Development Studies (HCPDS) programme website [www.haalsi.org]. Data are also accessible through the Inter-university Consortium for Political and Social Research (ICPSR) at the University of Michigan [www.icpsr.umich.edu] and the INDEPTH Data Repository [http://www.indepth-ishare.org/index.php/catalog/113]. Mortality data are available upon request.

**ORCID iDs**
Alisha N Wade http://orcid.org/0000-0002-1158-2523
Miles Witham http://orcid.org/0000-0002-1967-0990

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
