## [Reviewer comments · BMJ Open]

ARTICLE DETAILS

TITLE (PROVISIONAL)	Multimorbidity and mortality in an older, rural Black South African population cohort with high prevalence of HIV-findings from the HAALSI study
AUTHORS	Wade, Alisha N.; Payne, Collin F.; Berkman, Lisa; Chang, Angela; Gómez-Olivé, F. Xavier; Kabudula, Chodziwadziwa; Kahn, Kathleen; Salomon, Joshua; Tollman, Stephen; Witham, Miles; Davies, Justine

VERSION 1 – REVIEW

REVIEWER	Sabin, Caroline UCL Medical School
REVIEW RETURNED	26-Jan-2021

GENERAL COMMENTS	The authors of this study aimed to investigate the association of multi-morbidity with all-cause mortality in an established South African population-based cohort, and to investigate whether the association was different when HIV was one of the morbidities present compared to when it wasn't. Overall, the authors report that multi-morbidity is associated with an increased mortality risk, that frailty (also associated with mortality) modifies this risk slightly and that HIV itself is associated with increased mortality after adjustment. The authors incorrectly conclude that the latter association is only present in men – in fact, HIV appears to be associated with mortality in both men and women, however the association in women is clearly lacking in power and the finding is thus non-significant. This over-interpretation of a non-significant p-value concerns me as it suggests that the authors are simply running their analyses blindly without thinking about the biological (or other) mechanisms that explain these associations. Why would HIV be associated with mortality only in men? Whilst the results are of some interest, they unfortunately don't really add that much to what is already known about multi-morbidity and its association with mortality. My enthusiasm is dampened by three main factors. 1. Firstly, the authors considered a list of ten chronic morbidities, presumably because they already had the data on these. This list was a slightly odd list, including morbidities with varying prognoses, with no information on whether these were controlled by medication or not, nor the severity of each, and including events such as PTSD and alcohol dependence. In contrast, important morbidities including cancer, liver disease and renal disease were not included. No information was provided as to the relative frequency of each of these within those with multi-morbidity, nor whether the morbidities that were present differed
--

	between those with and without HIV, between men and women, between older and younger people etc. 2. The authors performed two sets of analyses – one on all-cause mortality using a Cox model, and the second considering two-year mortality with a logistic regression model. The models from each were broadly consistent, and I'm not sure that there is much to be gained by describing each set of analyses in detail. I would suggest the authors choose one of these to focus on in the main analyses and report the second analyses as part of supplementary material without describing this in detail in the main text. 3. The authors report that HIV was associated with poorer mortality – but without knowing if those with HIV had high/low CD4 counts, were on/off ART, had detectable or undetectable viraemia, etc., these findings don't help to understand why this was the case. And HIV status is likely to be confounded by other lifestyles or behaviours which are not captured through the single wealth index or the other factors in the model – whether the estimated risk of mortality due to HIV is totally explained by HIV itself or simply reflects this confounding should be discussed. Other specific comments: a. Frailty was defined using the Fried criteria, but this was adapted for several of the components so that individuals met the threshold for each criterion if the person's walking speed, say, fell in the lowest quintile of values, even if this was not a particularly low speed. Absolute low BMI was used in place of unintentional weight loss. These adaptations might artificially increase the proportion of people apparently being classified as frail and may limit any comparison to other studies. b. Results – there was no difference in age between those who could and couldn't be included due to missing data. The test used to generate the p-value wasn't clear here – was it a t-test or Mann-Whitney test? Those excluded were more likely to have frailty, which does contradict with the earlier assertion that the missing data was almost entirely due to problems with the measurement tools (which would argue for random missingness). Although this was mentioned in the Discussion, further attempts to investigate the robustness of findings to these missing data would have been helpful. c. Of the 459 who died, how many were men and how many were women? Further information on the characteristics of those who died, particularly in terms of the frailty, multi-morbidity and HIV status, would be helpful. d. Interpretation of AUC – the interpretation of this ('... there was a 76.7% chance that the model would discriminate between those who did and didn't die') feels a little odd – the alternative interpretation ('the model was able to correctly discriminate between those who did and didn't die in 76.7% of the pairs considered') is perhaps a more intuitive interpretation of the AUC although I appreciate that it boils down to the same thing. e. Table 3 – the row alignment seems a bit off at times for educational attainment and wealth quintile.
--	--

REVIEWER	Chivese, Tawanda University of Cape Town Faculty of Health Sciences, college of medicine
REVIEW RETURNED	01-Feb-2021

GENERAL COMMENTS	 1. Please state the mean age (SD) and gender in the abstract. 2. Please explain how multimorbidity was assessed - in the abstract 3. In the methods, please specify who did the initial translation and back translation of the questionnaire and their relationship with the study 4. In Table 1, please specify that age is reported as median and IQR if this is the case 5. Please present the numbers and proportions of participants who died in each of the key exposure groups
--

VERSION 1 – AUTHOR RESPONSE

Reviewer 1

1. Overall, the authors report that multi-morbidity is associated with an increased mortality risk, that frailty (also associated with mortality) modifies this risk slightly and that HIV itself is associated with increased mortality after adjustment. The authors incorrectly conclude that the latter association is only present in men – in fact, HIV appears to be associated with mortality in both men and women, however the association in women is clearly lacking in power and the finding is thus nonsignificant. This over-interpretation of a non-significant p-value concerns me as it suggests that the authors are simply running their analyses blindly without thinking about the biological (or other) mechanisms that explain these associations. Why would HIV be associated with mortality only in men?

We acknowledge that our lack of association between HIV multimorbidity and mortality in women may have been due to a lack of statistical power and have expanded our discussion on page 25 to include this.

We have also now reported differences in viral load between sexes which might also partially explain why there was a stronger association between HIV multimorbidity and mortality in men than in women (page 15) and included this point in our discussion (page 25).

2. Whilst the results are of some interest, they unfortunately don't really add that much to what is already known about multi-morbidity and its association with mortality. We respectfully disagree with this assertion by the reviewer. As we note in our introduction, virtually all of the studies on the relationship between multimorbidity and mortality have, to date, been conducted in high-income countries. We present one the first studies in a lower to middle income country with a high prevalence of HIV.

3. Firstly, the authors considered a list of ten chronic morbidities, presumably because they already had the data on these. This list was a slightly odd list, including morbidities with varying prognoses, with no information on whether these were controlled by medication or not, nor the severity of each, and including events such as PTSD and alcohol dependence. In contrast, important morbidities including cancer, liver disease and renal disease were not included. No information was provided as to the relative frequency of each of these within those with multi-morbidity, nor whether the morbidities that were present differed between those with and without HIV, between

men and women, between older and younger people etc.

As opposed to the case in high-income countries, electronic medical records often do not exist or are unreliable in lower- and middle-income countries. Hence, it is not usually possible to include the depth of information that the reviewer requests. That said, our data – collected by survey – does allow us to use the widely accepted Academy of Medical Sciences definition of multimorbidity, namely the co-existence of two or more chronic conditions. This conceptualisation permits assessment of the burden of multiple chronic conditions, regardless of what those chronic conditions may be and does not assign primacy to any particular condition. We do agree this definition has limitations as the severity and treatment status of individual conditions is not considered; we had acknowledged these limitations in our initially submitted manuscript and have now elaborated on them on pages 23-24. We had also included them in our summary of the strengths and limitations of the study. However, we have also now reported the relative frequency of the individual conditions within the definition of multimorbidity, overall and by sex, age group and HIV multimorbidity status on page 15 and in Supplementary table 1.

4. The authors performed two sets of analyses – one on all-cause mortality using a Cox model, and the second considering two-year mortality with a logistic regression model. The models from each were broadly consistent, and I'm not sure that there is much to be gained by describing each set of analyses in detail. I would suggest the authors choose one of these to focus on in the main analyses and report the second analyses as part of supplementary material without describing this in detail in the main text.

We have now presented the Cox model in the main analysis and the logistic regression as part of the supplementary material (Supplementary tables 2&3).

5. The authors report that HIV was associated with poorer mortality – but without knowing if those with HIV had high/low CD4 counts, were on/off ART, had detectable or undetectable viraemia, etc., these findings don't help to understand why this was the case. And HIV status is likely to be confounded by other lifestyles or behaviours which are not captured through the single wealth index or the other factors in the model – whether the estimated risk of mortality due to HIV is totally explained by HIV itself or simply reflects this confounding should be discussed.

We have now reported viral load in patients with HIV multimorbidity (page 15 and Table 1) and do note that only half of these individuals had a suppressed viral load. We have expanded our discussion to include this as well as to acknowledge other factors, not included in our model, which might confound the relationship between HIV multimorbidity and mortality (page 25).

6. Frailty was defined using the Fried criteria, but this was adapted for several of the components so that individuals met the threshold for each criterion if the person's walking speed, say, fell in the lowest quintile of values, even if this was not a particularly low speed. Absolute low BMI was used in place of unintentional weight loss. These adaptations might artificially increase the proportion of people apparently being classified as frail and may limit any comparison to other studies.

Our use of population-specific lowest quintiles to identify those with low walking speed, grip strength and physical activity is in keeping with Fried's approach. Additionally, our frailty criteria, used in this study, have demonstrated construct and predictive validity as shown in our previous publication (reference 16 in the manuscript). Both of these points have been clarified (pages 11, 24). However, our validated model did use an absolute body mass index as a proxy for unintentional weight loss, which we acknowledge may have inflated the prevalence of frailty in our study. We have now included this limitation in our discussion (page 24).

7. Results – there was no difference in age between those who could and couldn't be

included due to missing data. The test used to generate the p-value wasn't clear here – was it a t-test or Mann-Whitney test? Those excluded were more likely to have frailty, which does contradict with the earlier assertion that the missing data was almost entirely due to problems with the measurement tools (which would argue for random missingness).

Although this was mentioned in the Discussion, further attempts to investigate the robustness of findings to these missing data would have been helpful. We have clarified on page 12 that all continuous variables were compared using the Mann-Whitney test. We performed a sensitivity analysis in which the individual conditions with the highest degree of missingness were excluded from the definition of multimorbidity, with persistence of the relationships with mortality. This is reported on page 21 and in Supplementary table 4. We had also acknowledged in our original summary of the strengths and limitations of the study that the higher proportion of frailty in individuals who were excluded due to missing data may have diluted the impact of frailty on the relationship between multimorbidity and death.

8. Of the 459 who died, how many were men and how many were women? Further information on the characteristics of those who died, particularly in terms of the frailty, multi-morbidity and HIV status, would be helpful.

We have included further details on the individuals who died on page 15 and in Table 2.

9. Interpretation of AUC – the interpretation of this ('... there was a 76.7% chance that the model would discriminate between those who did and didn't die') feels a little odd – the alternative interpretation ('the model was able to correctly discriminate between those who did and didn't die in 76.7% of the pairs considered') is perhaps a more intuitive interpretation of the AUC although I appreciate that it boils down to the same thing. We have revised the wording in keeping with the reviewer's suggestion (pages 20-21).

10. Table 3 – the row alignment seems a bit off at times for educational attainment and wealth quintile.

This has been adjusted.

Reviewer: 2

1. Please state the mean age (SD) and gender in the abstract.

As age was not normally distributed, we have included the median age (interquartile range) in the abstract.

2. Please explain how multimorbidity was assessed - in the abstract.

This has been included.

3. In the methods, please specify who did the initial translation and back translation of the questionnaire and their relationship with the study.

This has been included on page 8.

4. In Table 1, please specify that age is reported as median and IQR if this is the case.

This has been included.

5. Please present the numbers and proportions of participants who died in each of the key exposure groups.

Details of the individuals who died have been included in Table 2.

VERSION 2 – REVIEW

REVIEWER	Sabin, Caroline UCL Medical School
REVIEW RETURNED	22-Jul-2021

GENERAL COMMENTS	The authors have now made substantial changes to their manuscript in response to my comments, and I think the manuscript is improved as a result. I have no further comments.
---

REVIEWER	Chivese, Tawanda University of Cape Town Faculty of Health Sciences, college of medicine
-----------------	---

REVIEW RETURNED	28-Jul-2021
-------------

GENERAL COMMENTS	The authors have addressed my concerns
--